# Biomarker-Based Assessment Model for Detecting Sepsis: A Retrospective Cohort Study

**DOI:** 10.3390/jpm13081195

**Published:** 2023-07-27

**Authors:** Bo Ra Yoon, Chang Hwan Seol, In Kyung Min, Min Su Park, Ji Eun Park, Kyung Soo Chung

**Affiliations:** 1Department of Internal Medicine, New Korea Hospital, Gimpo-si 10086, Republic of Korea; borayoon00@naver.com; 2Division of Pulmonology, Allergy and Critical Care Medicine, Department of Internal Medicine, Yongin Severance Hospital, Yonsei University College of Medicine, Yongin 16995, Republic of Korea; seolgoon@yuhs.ac; 3Biostatistics Collaboration Unit, Department of Biomedical Systems Informatics, Yonsei University College of Medicine, Seoul 03722, Republic of Korea; 4Department of Pulmonary and Critical Care Medicine, Ajou University School of Medicine, Suwon 16499, Republic of Korea; 5Division of Pulmonology, Department of Internal Medicine, Yonsei University College of Medicine, Seoul 03722, Republic of Korea

**Keywords:** septic shock, mortality, intensive care unit, quick sequential organ failure assessment, infection

## Abstract

The concept of the quick sequential organ failure assessment (qSOFA) simplifies sepsis detection, and the next SOFA should be analyzed subsequently to diagnose sepsis. However, it does not include the concept of suspected infection. Thus, we simply developed a biomarker-based assessment model for detecting sepsis (BADS). We retrospectively reviewed the electronic health records of patients admitted to the intensive care unit (ICU) of a 2000-bed university tertiary referral hospital in South Korea. A total of 989 patients were enrolled, with 77.4% (n = 765) of them having sepsis. The patients were divided into a ratio of 8:2 and assigned to a training and a validation set. We used logistic regression analysis and the Hosmer–Lemeshow test to derive the BADS and assess the model. BADS was developed by analyzing the variables and then assigning weights to the selected variables: mean arterial pressure, shock index, lactate, and procalcitonin. The area under the curve was 0.754, 0.615, 0.763, and 0.668 for BADS, qSOFA, SOFA, and acute physiology and chronic health evaluation (APACHE) II, respectively, showing that BADS is not inferior in sepsis prediction compared with SOFA. BADS could be a simple scoring method to detect sepsis in critically ill patients quickly at the bedside.

## 1. Introduction

‘Sepsis’ originated from the Greek word [σηψις], meaning ‘decomposition of animal or vegetable organic matter in the presence of bacteria’ more than 2700 years ago [1]. The current concept of sepsis was first introduced at the 1991 American College of Chest Physicians (ACCP)/Society of Critical Care Medicine (SCCM) Consensus Conference; it was recently redefined in the Third International Consensus Definitions for Sepsis and Septic Shock (Sepsis-3) as ‘life-threatening organ dysfunction caused by a dysregulated host response to infection’ [2,3]. Although sepsis is a significant cause of death in critically ill patients, because of this conceptual ambiguity, providing clinically apparent diagnostic criteria and early detection is difficult.

Since the 1980s, several mortality prediction scoring systems have been proposed for intensive care unit (ICU) patients; however, these are not suitable for sepsis diagnosis [4,5,6,7]. The sequential organ failure assessment (SOFA) was introduced by the European Society of Intensive Care Medicine (ESICM) in 1994 to assess the severity of organ dysfunction in critically ill patients. Given that multi-organ dysfunction (MOD) has been identified as a major cause of death in ICU patients, SOFA has been used for a long time as a predictor of mortality [5]. However, the causes of MOD are diverse, and a high SOFA score does not always mean severe sepsis. Moreover, because of the need for laboratory findings and physical examinations, SOFA could not be easily and quickly applied in sepsis patients, where rapid diagnosis is essential. To overcome these shortcomings, Sepsis-3 introduced a new concept of quick SOFA (qSOFA) as a simple bedside score, which can be easily applied compared to the SOFA score [3].

It is indisputably accepted that qSOFA can detect sepsis using only vital signs and mental states. However, despite that, qSOFA was developed for sepsis detection, while SOFA was developed for the prediction of MOD and mortality; the variables are still not pathognomonic in sepsis, and they could be affected by causes other than sepsis. Moreover, in the emergency department, where early detection of sepsis is also essential, there are some discussions about whether qSOFA alone can predict sepsis and mortality [8,9,10]. qSOFA and SOFA do not contain the concept of suspected infection, which is still ambiguous in sepsis diagnosis. Therefore, physicians should work particularly hard to find suspected infections in patients with higher qSOFA or SOFA scores. Unfortunately, the calculation of sequential SOFA is complicated and takes time, which makes it less feasible to be performed directly at the patient’s bedside.

The generalization of electronic medical records (EMR) and advancements in artificial intelligence (AI) have facilitated the investigation of sepsis diagnosis through machine learning [11,12]. A recent study attempts to diagnose sepsis using unstructured clinical data, such as clinical text and images, along with vital signs and laboratory data [13]. However, there are still many hurdles on the path of sepsis diagnosis through AI technologies being universally used worldwide in the medical field. In particular, it cannot be used in countries or medical institutions where electronic medical record systems are poorly established. Rather, the medical field needs a tool, which can intuitively identify patients with sepsis at the bedside.

Focusing on these points, we aimed to introduce our newly developed simple biomarker-based assessment model for detecting sepsis (BADS). We took into consideration the following aspects: (1) the model should include the variables for infections, and subsequent steps should not be required; (2) the model can be applied simply in ICU patients; and (3) the model can help predict multi-organ failure or mortality similarly to other methods.

## 2. Materials and Methods

### 2.1. Study Design and Patient Population

This study was a single-center retrospective cohort study, and the study protocol was approved by the institutional review board (IRB) of the Severance Hospital (IRB number: 4-2017-0654) and performed following the Helsinki Declaration. Individual patient data were stored as an encrypted file, and only authorized investigators were able to access personal medical records to protect the privacy of patients. The IRB committee approved the waiver of informed consent because the study was retrospective in design, and the personal information of patients enrolled in this study was removed from all sections of the manuscript by the HIPAA privacy rule.

We analyzed a total of 989 patients aged >18 years who were admitted to the medical ICU at the Yonsei University Severance Hospital from March 2015 to September 2017. Patients who were re-admitted during the period were the only ones excluded. Patients were divided into a ratio of 8:2 by random sampling, and they were allocated to a training and a validation set. A new diagnosis model was developed using the ‘training set’, and the diagnostic performance of the model was evaluated in the validation set (Appendix A).

### 2.2. Variables and Definitions

Two intensivists retrospectively categorized ICU patients into sepsis and non-sepsis using information from the hospital records, including patients’ history, clinical presentation, microbiological results, and discharge administrative claim codes (international statistical classification of disease and related health problems—10 codes for sepsis (A02.1, A20.7, A22.7, A24.1, A26.7, A32.7, A39.2, A39.3, A39.4, A40.x, A41.x, A42.7, B00.7, B37.7, O85, P36.x), severe sepsis (R65.1), septic shock (R57.2)). According to the Sepsis-3 diagnostic criteria, sepsis was diagnosed when a patient with a suspected or confirmed infection had an acute change of 2 or more points in the SOFA score. This change was directly associated with the infection, leading to a diagnosis of sepsis. However, if there was no suspected infection related to sepsis in the judgment of the critical care specialists who reviewed the data retrospectively, the case was deemed not to be sepsis, even if the SOFA score had changed by 2 points or more.

The prediction power of the new model was compared to that of the SOFA, qSOFA, and acute physiology and chronic health evaluation II (APACHE II) [4,5]. In the Sepsis-3 definition, sepsis is suspected when the patient has a suspected infection, and the qSOFA score is accompanied by two or more of the following conditions: incorporating altered mentation, systolic blood pressure ≤100 mm Hg, and respiratory rate ≥22 breaths/min [3].

### 2.3. Data Collection and Clinical Outcomes

We collected the data regarding the baseline demographics, pre-existing comorbidities, and other laboratory findings obtained within 24 h of ICU admission. At the same time, the patients were assessed for whether they had non-sepsis or sepsis at the time of ICU admission by reviewing the discharge administrative claim codes.

The goal of the new scoring system was to predict sepsis probability. As the primary outcome, sepsis probability was calculated according to the total score. After determining the cut-off level of the new model, we divided patients into the low- and high-score groups and compared the 28-day overall mortality between the two groups as the secondary outcome.

### 2.4. Statistical Analysis

The clinical parameters were analyzed using Student’s *t*-test or the Mann–Whitney U-test for continuous variables and the chi-squared test or Fisher’s exact test for categorical variables. Categorical variables were expressed as numbers with percentages and continuous variables as medians with interquartile ranges.

In the training set, multivariable logistic regression was used to compare the contribution of variables. Variables identified as highly correlated with sepsis diagnosis, which could be verified from the patient’s bedside, were selected. These items assigned weighted proportional points based on their β-coefficients to develop the new prediction model (called the ‘biomarker-based assessment model for detecting sepsis’ (BADS)). The suitability of the model was confirmed through the Hosmer–Lemeshow test.

The total BADS score as sepsis probability was calculated for each patient, and the discrimination for sepsis was evaluated using the area under the receiver operating characteristic (AUROC) curve. According to the cut-off score determined by the ROC curve, the patients were divided into low- and high-score groups, and the mortality of both groups was compared by using the Kaplan–Meier curve.

In the training set, we displayed a calibration plot using the bootstrap method and examined the model’s suitability using the Hosmer–Lemeshow test. ROC comparison analysis was used to compare the prediction power of BADS with that of the SOFA, APACHE II, and qSOFA scores. Cross-validation was performed, and the integrated results were calculated to generalize the results. A two-sided *p* value < 0.05 was considered statistically significant. Statistical analyses were performed using R statistical software, version 3.4.1 (The R Foundation for Statistical Computing, Vienna, Austria).

## 3. Results

### 3.1. Baseline Characteristics of the Study Population

A total of 989 patients were enrolled in this study, with sepsis incidence among 77.4% (n = 765). Patient baseline demographics are described in Table 1. Compared to the non-sepsis group, there were more older patients in the sepsis group (68 vs. 64 years, *p* = 0.034), and the severity at the time of admission was higher in this group: SOFA (10 vs. 6, *p* < 0.001), qSOFA (2 vs. 1, *p* < 0.001), and APACHE II (25 vs. 19, *p* < 0.001). There was no statistically significant difference in the Charlson comorbidity index (3 vs. 3, *p* = 0.274). Most of the patients showing positive results in blood culture were diagnosed as having sepsis, except for a few cases with non-sepsis, who were finally identified as contamination (35.7% vs. 8.0%, *p* < 0.001). The sepsis group had worse clinical outcomes, which were as follows: 28-day mortality (34.8% vs. 18.8%, *p* < 0.001), ICU mortality (32.9% vs. 15.2%, *p* < 0.001), and hospital mortality (50.8% vs. 38.4%, *p* = 0.001).

Other clinical parameters and findings were also compared, as shown in Table 2. The following hemodynamic variables were found more unstable in the sepsis group than in the non-sepsis group: MAP (66 vs. 78 mmHg, *p* < 0.001), heart rate (106 vs. 95 beats/min, *p* < 0.001), and shock index (1.2 vs. 1.0, *p* < 0.001). The parameters indicating severe infection were worse in the sepsis group: white blood cell (13.3 × 10^3^/μL vs. 8.5 × 10^3^/μL, *p* < 0.001), delta neutrophil index (4.0% vs. 1.5%, *p* < 0.001), lactate (2.4 vs. 1.3 mmol/L, *p* < 0.001), C-reactive protein (102.9 vs. 44.9 mg/L, *p* < 0.001), and procalcitonin (1.80 vs. 0.50 ng/mL, *p* < 0.001); the platelet count (121 × 10^3^/μL vs. 151 × 10^3^/μL, *p* = 0.001) was lower in these patients. Each clinical characteristic of the training and validation cohorts is presented in Appendix A.

### 3.2. Development of the New Sepsis Prediction Model

The contribution of each variable and the odds ratios of the four selected variables are described in Table 3. The contributions of the four variables were analyzed by logistic regression analysis (Appendix A).

The BADS score was developed using the following formula (Figure 1):1/(1 + exp(−η)), where η = 1.3805 − 0.0147 × MAP + 0.0727 × procalcitonin + 0.0909 × lactate + 0.4223 × shock index

In the training set, the sepsis discrimination power of BADS was analyzed (AUC, 0.766; 95% confidence interval [CI], 0.724–0.809), and the goodness of fit in the logistic regression model was evaluated (*p* = 0.752). This result was also statistically significant in the validation set: sepsis discrimination (AUC, 0.668; 95% CI, 0.598–0.777) and goodness of fit in the logistic regression model (*p* = 0.089) (Figure 2).

The prediction power of BADS was compared with that of qSOFA, SOFA, and APACHE II and generalized through cross-validation (Figure 3). The average AUC was 0.754, 0.615, 0.763, and 0.668 for BADS, qSOFA, SOFA, and APACHE II, respectively. The AUC of BADS was higher than that of qSOFA (AUC difference, 0.139; 95% CI, 0.090–0.188) and APACHE II (AUC difference, 0.087; 95% CI, 0.034–0.143), and the difference was statistically significant for qSOFA. Compared with the SOFA, the prediction power of BADS was low but not statistically significant (AUC difference, −0.009; 95% CI, −0.055–0.035).

### 3.3. Predictive Performance for Mortality in Sepsis

We set the cut-off score of sepsis diagnosis as 11.4802 (sensitivity, 0.634; specificity, 0.765). Then, the mortality was compared between the low- and high-score groups based on this cut-off score (Figure 4). As the total points of BADS increased, the 28-day overall mortality also increased, and this result was statistically significant (hazard ratio, 2.282; 95% CI, 1.744–2.986; log rank *p* < 0.001) (Figure 4).

## 4. Discussion

We developed a new scoring system, which can be simply used in ICU patients at the bedside. Compared with the SOFA score, BADS was not statistically inferior concerning sepsis discrimination, and a statistically significant increase in mortality was observed in the high-score group.

According to the discharge data, the incidence and mortality associated with sepsis are still increasing [14]. In the Sepsis-3 definitions, qSOFA was introduced to help in the detection of sepsis in patients with suspected infections [3]. However, the variables, such as tachypnea, hypotension, and unconscious mental state, are not pathognomonic changes, which are only caused by sepsis, and qSOFA does not include the concept of infection. Moreover, in the initial approach for patients in the emergency department, qSOFA alone was reported to be clinically ineffective in predicting sepsis and mortality. Thus, additional validations of qSOFA are required because it does not replace existing early warning scores, even in retrospective studies [8,9,10,15].

It is not easy to identify infections or suspected infections, since the source of infection and causative organisms are not always obvious, and positive cultures may not reflect the actual infections. Moreover, the concept of sepsis is ambiguous; thus, it is not easy to identify one or two diagnostic variables or develop models, which can predict sepsis. In this study, we finally selected four sepsis-related variables, which could be measured in a simple way.

The respiratory rate is known to be an early indicator of patient deterioration in both adults and children, and it is often the activation criterion for the rapid response team [16,17,18]. Blood pressure, pulse rate, temperature, and oxygen saturation are routinely measured in an automated manner; however, the respiratory rate remains the only parameter, which is measured manually, and there is no precise way to measure the respiratory rate accurately [19]. In several retrospective studies, the inaccuracy in measuring the respiratory rate was described. The values tend to be clustered at 18 and 20 breaths/minute, as observers usually multiply the values measured for a shorter period instead of measuring the respiratory rate for a full minute, even though respiratory rates are not always regular. Moreover, patients who are aware that the respiratory rate is being measured may affect the results [20,21]. Thus, further randomized controlled trials or prospective studies for the accurate measurement of the respiratory rate may be required to increase the reliability.

Conversely, in the Surviving Sepsis Campaign Guidelines, a MAP of more than 65 mmHg is usually recommended as an indicator for adequate fluid resuscitation in patients with sepsis [22,23]. Mean arterial pressure is often used as an index for maintaining renal perfusion or adjusting vasopressor capacity [24]. The shock index, defined as the ratio of heart rate to systolic arterial pressure, is also known to be more useful in assessing hemodynamic stability than the conventional vital signs in critically ill patients [25,26,27]. However, the pulse rate can be influenced by vasopressors and other factors, which limits the accuracy of the shock index. Thus, these two clinical indicators, which can be measured at the bedside, have different clinical meanings, and we selected the MAP and shock index as indicators of circulation failure and organ hypoperfusion due to septic shock.

However, clinically, hypoperfusion does not always result in gross changes in blood pressure or heart rate. Sometimes, the patient is normotensive, but organ perfusion is still insufficient. MAP and the shock index are more representative of severe sepsis or septic shock than traditional markers, but they do not reflect occult hypoperfusion. Lactate is known to be increased as a result of anaerobic glycolysis due to inadequate oxygen delivery in septic shock [28,29]. Several studies have shown that the lactate level is more sensitive to tissue hypoxia compared to other parameters, and it is associated with higher mortality if not normalized by treatment [30,31,32,33]. Therefore, lactate was also used in our model to complement the concealed hypoxic changes through sepsis.

Procalcitonin was first described as a possible biomarker of sepsis in 1993 [34]. Since then, it is known to increase with the severity of bacterial infection, and unlike C-reactive protein, it is rarely affected by the glucocorticoid level. Thus, procalcitonin has been widely used in sepsis diagnosis and antibiotic use guidelines [34,35,36]. There are many other biomarkers with higher sensitivity than procalcitonin; however, it takes a relatively long time to obtain the results of these markers. Procalcitonin, similar to lactate, can be quickly identified through point-of-care testing (POCT), and recently, the use of handheld mobile devices has also been introduced [37,38,39]. On the one hand, using point-of-care testing equipment in institutions or countries lacking diagnostic capabilities can enable procalcitonin or lactate measurement, making it more accessible in low- and middle-income countries. Therefore, procalcitonin and lactate, which can be assessed at the bedside, are the most appropriate biomarkers for detecting sepsis in cases where rapid diagnosis is a crucial factor for prognosis.

The strength of BADS is that it is a model developed explicitly for sepsis. Moreover, it was not inferior in sepsis prediction compared with the SOFA score, and it was as simple as qSOFA.

However, this study also has several limitations. First, this study was performed retrospectively in a single center, and thus, the findings presented here require additional validation in the other forms of ICU and well-organized studies, such as randomized controlled prospective investigations. Second, although it is theoretically possible, this score has not been applied to non-ICU patients yet. Finally, the variables we chose for the model have their limitations. The Surviving Sepsis Campaign Guidelines recommended a MAP target of 65 mmHg in patients with evidence of septic shock; however, this value may not always be acceptable. The perfusion pressure should be applied differently based on the individual patient’s response or organs; for example, when chronic hypertension is present, 80–85 mmHg is more reasonable [40]. The shock index is useful for the assessment of hemodynamic stability by hypovolemia because it is relatively simple and accurate for evaluating left ventricular dysfunction and circulatory failure [25,27]. However, if the acute volume loss is insufficient, the increase in the ratio is not definite, and the response may be inaccurate when the patients have an arrhythmia or use drugs, which affect the heart rate, such as beta-blockers or calcium channel blockers [41,42]. Hyperlactatemia can occur not only through overproduction but also through underutilization. When liver and kidney dysfunction is present, the lactate level may also be increased regardless of hypoxia. Overproduction of lactate resulting from glycolysis in the absence of oxygen, classified as type A lactic acidosis, may also be caused by conditions other than septic shock. Lung lactate is increased in acute lung injury, and hemoglobin transfer disorder or circulatory failure due to other causes may also result in type A lactic acidosis [43,44,45]. However, regardless of the cause of hyperlactatemia, lactate elevation is revealed to be associated with a worse outcome, and several guidelines suggest lactate-guided resuscitation to improve clinical outcomes [23,28,46]. Serum procalcitonin concentration, although proved to be increased by infections, was first identified as a precursor of calcitonin, which is normally produced from C-cells of the thyroid. Therefore, procalcitonin production may increase for various reasons other than infections. For example, it may be produced due to paraneoplastic syndromes, treatment with agents that stimulate cytokines, and other stressful conditions, such as burns, surgery, and trauma, among others. Conversely, in the case of infection by fungus or virus, the increase in procalcitonin is not as prominent as that in bacterial infection despite sepsis [35,36,47]. Therefore, the BADS score, which was combined for each variable, should be prospectively validated to determine whether the limitations of these variables will affect the diagnosis of sepsis in actual clinical practice.

Clinically, sepsis presents various manifestations; thus, accurate diagnosis of sepsis using one or two variables or scores is difficult. In this study, we developed a simple but more specific model for sepsis to help diagnose sepsis and predict mortality. Based on our findings, we can carefully assume that the probability of sepsis and 28-day overall mortality will increase with a high BADS score. Further research is needed for external validation and applicability in other clinical situations. Sepsis can occur in various cases, and it is necessary to verify its effectiveness in non-medical ICU settings, especially in general wards or emergency departments.

## 5. Conclusions

The BADS score is a simple scoring method, which can detect sepsis more rapidly in critically ill patients and may be helpful in complementing the other scoring systems in the prediction of mortality.

## Figures and Tables

**Figure 1 jpm-13-01195-f001:**
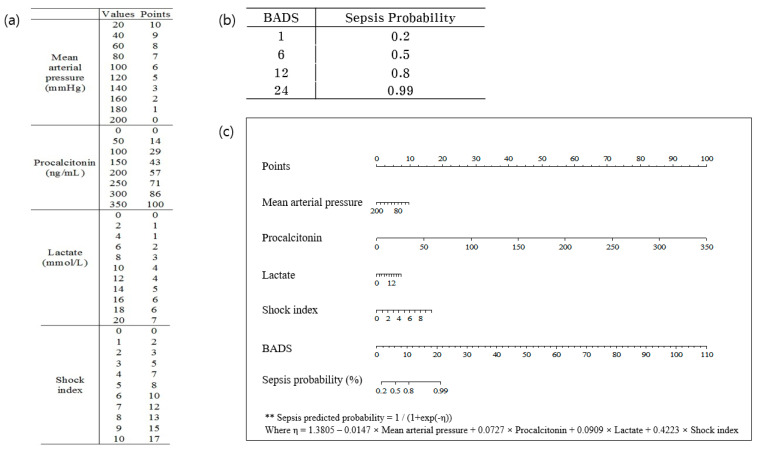
Nomogram of BADS score. Four selected variables (mean arterial pressure (MAP), shock index, lactate, and procalcitonin) and calculated probability of sepsis using a biomarker-based assessment model for detecting sepsis score (BADS). (**a**) The points of each variable after assigning the weight, (**b**) Probability of sepsis according to total score, (**c**) Nomogram of the scoring system. BADS, biomarker-based assessment model for detecting sepsis. ** is the formula for the sepsis predicted probability used in the nomogram.

**Figure 2 jpm-13-01195-f002:**
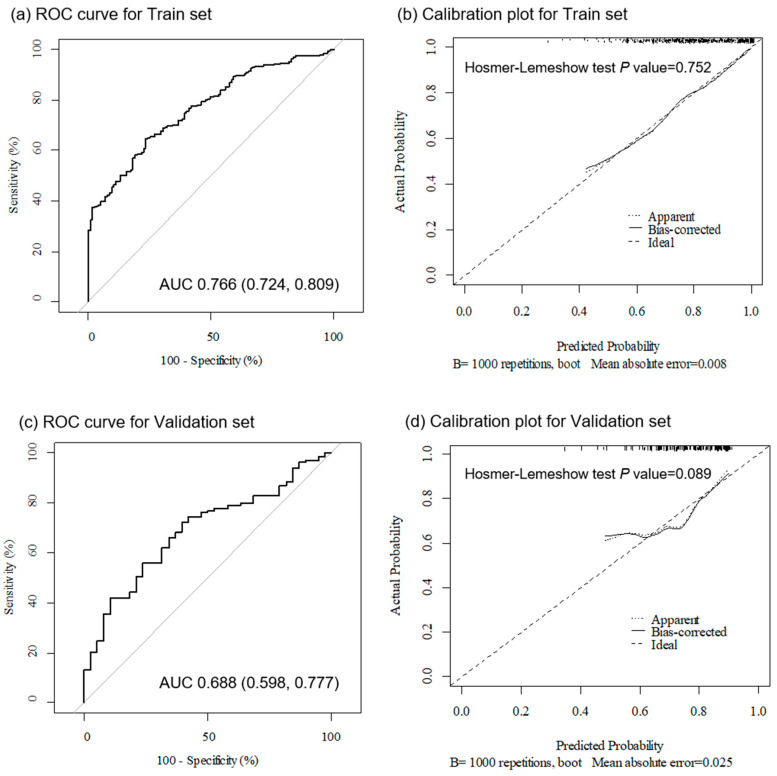
The predictive value and internal validation of sepsis prediction score. ROC curves for sepsis diagnosis with random prediction line in (**a**) training and (**c**) validation cohorts. Model calibration results for sepsis diagnosis in (**b**) training and (**d**) validation cohorts. ROC, receiver operating characteristic; AUC, area under the curve.

**Figure 3 jpm-13-01195-f003:**
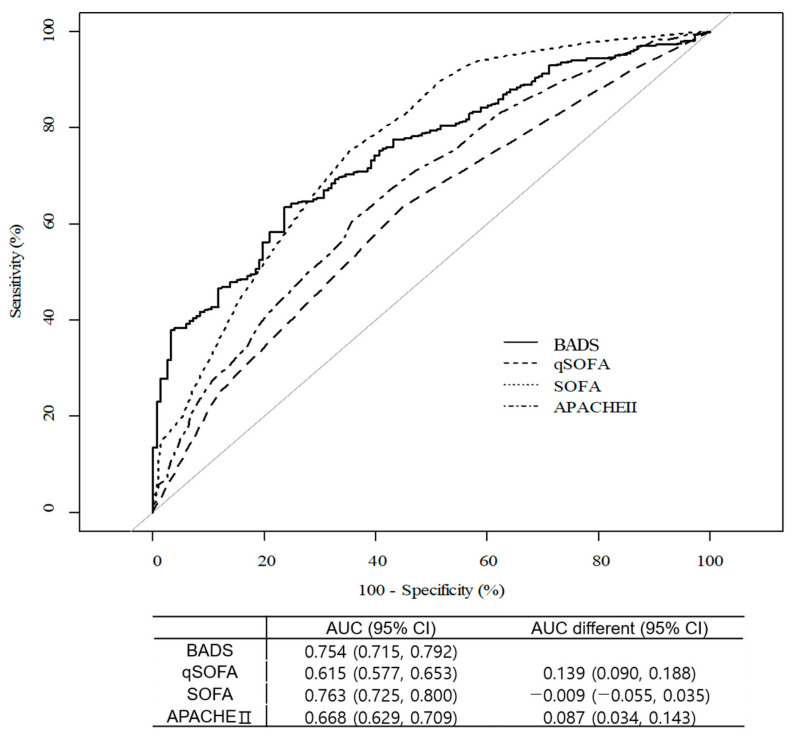
Comparison of sepsis prediction score for qSOFA, SOFA, and APACHE II. The AUC difference represents calculated values compared to BADS score. qSOFA, Quick Sequential Organ Failure Assessment; SOFA, Sequential Organ Failure Assessment; APACHE II, Acute Physiology and Chronic Health Evaluation Score II; AUC, area under the curve; BADS, biomarker-based assessment model for detecting sepsis.

**Figure 4 jpm-13-01195-f004:**
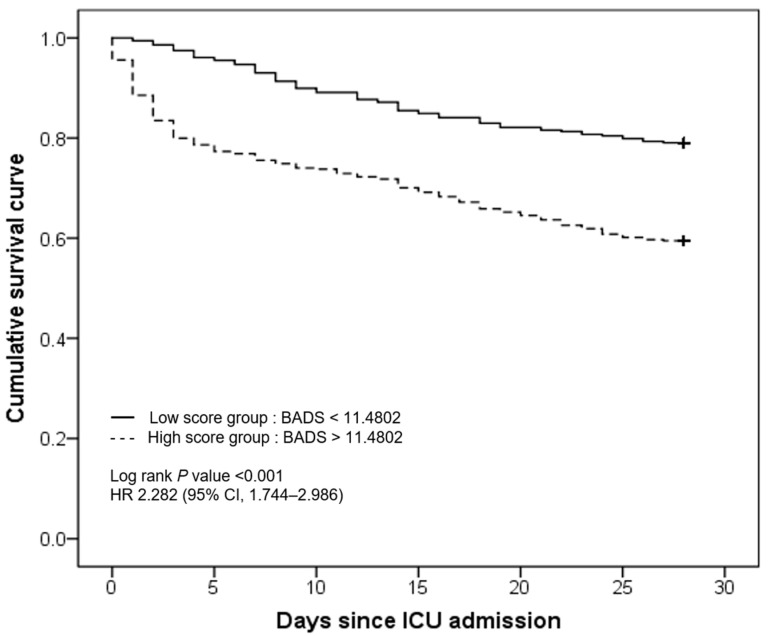
Twenty-eight-day mortality of patients according to high- and low-score group according to the sepsis diagnostic cut-off score (11.4802). The mortality was significantly increased in the high-score group. BADS, biomarker-based assessment model for detecting sepsis.

**Table 1 jpm-13-01195-t001:** Baseline characteristics of patients admitted to medical ICU.

	Non-Sepsis	Sepsis	*p* Value
	n = 224 (22.6%)	n = 765 (77.4%)	
Age (year)	64 (56, 75)	68 (57, 77)	0.034
Sex (male), n (%)	132 (58.9)	487 (63.7)	0.198
BMI (kg/m^2^)	22.2 (19.4, 24.3)	22.0 (19.5, 24.8)	0.905
qSOFA score	1 (1, 2)	2 (1, 2)	<0.001
0 points, n (%)	31 (13.8)	60 (7.8)	
1 point, n (%)	92 (41.1)	216 (28.2)	
2 points, n (%)	75 (33.5)	303 (39.6)	
3 points, n (%)	26 (11.6)	186 (24.3)	
SOFA score	6 (3, 9)	10 (7, 13)	<0.001
APACHE II score	19 (14, 26)	25 (19, 32)	<0.001
CCI	3 (2, 5)	3 (1, 4)	0.274
Comorbidity disease, n (%)			
Congestive heart failure	39 (17.4)	76 (9.9)	0.002
Coronary arterial disease	28 (12.5)	104 (13.6)	0.672
Chronic pulmonary disease	45 (20.1)	116 (15.2)	0.079
Chronic kidney disease	53 (23.7)	136 (17.8)	0.049
Chronic liver disease	22 (9.8)	79 (10.3)	0.802
Cerebrovascular disease	28 (12.5)	132 (17.3)	0.089
Solid cancer	55 (24.6)	227 (29.7)	0.136
Hematologic malignancy	8 (3.6)	52 (6.8)	0.075
ARDS, n (%)	13 (5.8)	86 (11.2)	0.017
PaO_2_/FiO_2_ ratio	232.8 (148.9, 360.1)	190.0 (113.2, 296.4)	<0.001
AKI, n (%)	54 (24.1)	254 (33.2)	0.010
Positive blood culture, n (%)	18 (8.0)	273 (35.7)	<0.001
28-day mortality, (non-survivors), n (%)	42 (18.8)	266 (34.8)	<0.001
ICU mortality, (non-survivors), n (%)	34 (15.2)	252 (32.9)	<0.001
Hospital mortality, (non-survivors), n (%)	86 (38.4)	389 (50.8)	0.001

Values are expressed as n (%) or median (interquartile range) unless otherwise indicated. BMI, Body Mass Index; qSOFA, Quick Sequential Organ Failure Assessment; SOFA, Sequential Organ Failure Assessment; APACHE II, Acute Physiology and Chronic Health Evaluation Score III; CCI, Charlson Comorbidity Index; ARDS, Acute Respiratory Distress Syndrome; PaO_2_/FiO_2_ ratio, The ratio of partial pressure arterial oxygen and fraction of inspired oxygen; AKI, Acute Kidney Injury.

**Table 2 jpm-13-01195-t002:** Clinical parameters of patients at medical ICU admission.

	Non-Sepsis	Sepsis	*p* Value
	n = 224 (22.6%)	n = 765 (77.4%)	
Mean arterial pressure (mmHg)	78 (66, 101)	66 (55, 80)	<0.001
Heart rate (beats/min)	95 (79, 112)	106 (88, 126)	<0.001
Shock index	1.0 (0.7, 1.3)	1.2 (0.9, 1.6)	<0.001
WBC (10^3^/μL)	8.5 (5.6, 13.1)	13.3 (7.2, 20.2)	<0.001
Hct (%)	28.3 (24.2, 33.0)	28.3 (24.5, 33.2)	0.997
RDW (%)	15.2 (14.0, 16.8)	15.4 (14.1, 17.2)	0.236
Platelet (10^3^/μL)	151 (83, 221)	121 (59, 210)	0.001
DNI (%)	1.5 (0.2, 3.9)	4.0 (1.6, 13.1)	<0.001
BUN, (mg/dL)	27.5 (17.2, 54.0)	33.1 (20.3, 50.8)	0.036
Creatinine (mg/dL)	1.0 (0.7, 2.7)	1.4 (0.8, 2.6)	0.046
Albumin (g/dL)	2.7 (2.3, 3.1)	2.5 (2.2, 2.9)	<0.001
Total bilirubin (mg/dL)	0.6 (0.4, 1.1)	0.8 (0.5, 1.7)	<0.001
Sodium (mmol/L)	137 (134, 141)	138 (134, 142)	0.024
Potassium (mmol/L)	4.2 (3.4, 4.7)	3.9 (3.3, 4.7)	0.147
Lactate, (mmol/L)	1.3 (0.9, 2.2)	2.4 (1.5, 5.0)	<0.001
CRP (mg/L)	44.9 (15.8, 108.6)	102.9 (42.3, 181.7)	<0.001
Procalcitonin (ng/mL)	0.50 (0.20, 1.80)	1.80 (0.40, 14.05)	<0.001

Values are expressed as n (%) or median (interquartile range) unless otherwise indicated. GCS score, Glasgow Coma Scale score; WBC, White Blood Cell; Hct, Hematocrit; RDW, Red Cell Distribution Width; DNI, Delta Neutrophil Index; BUN, Blood Urea Nitrogen; CRP, C-Reactive Protein.

**Table 3 jpm-13-01195-t003:** Logistic regression analysis of variables for sepsis diagnosis (sepsis vs. non-sepsis).

	Univariable Logistic			Multivariable Logistic		
	OR	95% CI	*p* Value	OR	95% CI	*p* Value
Mean arterial pressure	0.982	0.977–0.988	<0.001	0.985	0.979–0.992	<0.001
Procalcitonin	1.094	1.054–1.136	<0.001	1.075	1.038–1.114	<0.001
Lactate	1.177	1.101–1.258	<0.001	1.095	1.028–1.166	0.005
Shock index	2.384	1.735–3.274	<0.001	1.526	1.075–2.165	0.018

The effects of the variables on sepsis are described as odds ratio, 95% confidence interval, and *p* value. OR, Odds Ratio; CI, Confidence Interval.

## Data Availability

The datasets generated and/or analyzed in the current study are not publicly available due to the ‘Personal Information Protection Act’ law. However, anonymized datasets are available from the corresponding author upon reasonable request and with the consent of the IRB.

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
