# Peer review of "Biomarker-Based Assessment Model for Detecting Sepsis: A Retrospective Cohort Study"

_jpm, 2023, doi:10.3390/jpm13081195_

Round 1

Reviewer 1 Report

Biomarker-based assessment model for detecting sepsis: A ret-2 prospective cohort study (Manuscript ID: 2462528)

This study screened out the laboratory indicators related to the prognosis of sepsis through univariate and multivariate regression analysis, combined with mean arterial pressure and shock index, and obtained a new scoring model to predict the prognosis of sepsis, which has certain significance. However, the conclusions are not solid and convincing enough, and the manuscript is of lower quality. publication is not recommended.

Reason as follow:

1.      Lack of innovation

2.      In line 85, Why the author chooses 8:2 instead of 1:1, needs to be explained clearly

3.      At present, there are many prediction models related to the prognosis of sepsis, which the author did not mention in the introduction part of this article, and a background introduction is needed.

4.      In line 115, this article has done a 28-day death prognosis analysis, whether to fit the model in this article with the 60-day, 90-day, 180-day and 360-day prognosis.

5.      In line 181, the AUC area of ​​BADS is 0.766, which is less than 0.8. Overall, the reliability of this model is not high, and its practical application is limited.

6.      Table 1 Non-sepsis group, SOFA score (3,9), ICU patients are very prone to infection during hospitalization, how to rule out that these patients with SOFA>2 are not sepsis?

7.      Table 3 Because the mean arterial pressure and the shock index are still statistically significant in the multivariate analysis, the author included the mean arterial pressure and the shock index into the final model, but these two indicators have many influencing factors. The mean arterial pressure is related to the treatment of ICU doctors, and changes greatly. If timely fluid resuscitation and early use of vasoactive drugs, the mean arterial pressure can be high, even in shock patients. Similarly, the shock index has a lot of human influencing factors. These two points can actually be attributed to one point, whether the patient has septic shock or not, there is no need to put the two inside one model.

8.      Multi-factor regression analysis, including too many meanless factors, often obscures more important indicators. For example, age is closely related to prognosis.

9.      The weight of PCT in Figure 2 Nomogram is too large, with a maximum of 100 points, while the weight of mean arterial pressure and lactic acid level is very low. In actual clinical work, the level of PCT is not closely related to the severity of sepsis. In the case of gram-positive bacterial infection, fungal infection, intracranial infection, etc., the PCT may not be high, but the condition is also very serious. The weight of PCT in the Nomogram will seriously affect the accuracy of the model, which is questionable.

10.    The literature of the article is relatively old, update the literature.

11.    The author can choose ICU patients for 3-5 years after 2017 and make a verification database, so that the article will much more convincing.

12.    Optimize the model combination to make the model more meaningful, instead of simply incorporating the positive data of multivariate analysis to combine the model, more reference to the current sepsis model.

13.    Part of the language needs further polishing.

 Part of the language needs further polishing.

Reviewer 2 Report

Dear Editor

Thank you for the oppertunity to review the paper by Yoon et al.

The authors could benefit from adhering to the TRIPOD guidelines for reporting of prediction models.

I have som key methodological issues that needs to be adressed and clarified before i can recommend publication of the present paper:

1) Timing of exposure and outcome: It is stated that the outcome is sepsis on ICU admission, but that values are recorded during the first 24 hours of ICU stay - i.e. exposure/predicition variables are recorded after outcome which of course does not make sense

2) The outcome definition "sepsis" is difficult to reproduce as this is based on the clinical assesment of doctors whitout any specification of what variables / definitions this is based on and i.e. leads to 10% of non-sepsis to have bloodculture positiv bacterimia. Please be more concised and reproducible. 

3) The interpretation of the AUC value is overly optimistic.

4) The clinical characteristics of develoment vs. validation cohorts should be presented. 

5) It is concluded that the model is fast - how does is this supported by the data?

6) Please specify the process of variable selection in substantially greater detail

English language ok

Reviewer 3 Report

The article is interesting, and proposes a new algorithm for the diagnosis of sepsis upon hospitalization of the patient.

1) The results demonstrated that the proposed algorithm has a reliability similar to the SOFA score.

The Authors write in lines 66-67 "" sequential SOFA calculation is too complex and time consuming to be directly applied at the bedside". This is not entirely true. The SOFA score is based on biochemical indices (platelet count, bilirubin, creatinine, P/F) that are part of urgent emergency room determinations, and the cardiovascular score can be easily entered into a mobile app.

Instead, I wonder if the dosage of procalcitonin is so easy to obtain quickly in the emergency room.

2) regarding procalcitonin, I would ask for a comment on the costs. Wouldn't it be more useful to add the lactate dosage, available with blood gas analysis, to the classic SOFA score?

good

Reviewer 4 Report

The use of score systems in the diagnostic workflow of sepsis has been  debated by many Authors over time. 

In this paper Authors present results of a simple scoring system in a wide population of patients with suspected sepsis. 

There are some limitations of the study that should be discussed. Mainly, the retrospective approach and the assignement of patients to sepsis or not sepsis group may introduce a potential bias. Differences in accuracy  between train set and validation set should also be addressed.

No revision needed.

Round 2

Reviewer 1 Report

The results of this report exhibits that BADS is not inferior in sepsis prediction 30 compared with SOFA. BADS could be a simple scoring method to detect sepsis in critically ill patients quickly at the bedside.  In sepsis research field, many biomarkers have been detected, and the relationship or the mechanisms involved in sepsis were demonstrated. However, SOFA, qSOFA and APACHE scoring systems even have advantages over these biomarkers.  These biomarkers could be used with scoring systems to evaluate sepsis degree. 

English writing should be polished and edited.

Author Response

July 2023

Editor, Journal of personalized medicine

Manuscript ID: jpm-2462528

Manuscript Title: Biomarker-based assessment model for detecting sepsis: A retrospective cohort study

We want to thank all of the editors and reviewers for helping us make a better revision. We revised our manuscript according to the comments and recommendations of the reviewers. Here we include a separate itemized series of responses to the comments of the reviewers.

---------------------------------------------------------------------------------------------------------------------------

Reviewer #1

  • The results of this report exhibits that BADS is not inferior in sepsis prediction 30 compared with SOFA. BADS could be a simple scoring method to detect sepsis in critically ill patients quickly at the bedside.  In sepsis research field, many biomarkers have been detected, and the relationship or the mechanisms involved in sepsis were demonstrated. However, SOFA, qSOFA and APACHE scoring systems even have advantages over these biomarkers.  These biomarkers could be used with scoring systems to evaluate sepsis degree. 

Response: Thank you for your thoughtful comments. There are hundreds of biomarkers associated with the pathophysiology of sepsis. However, only a few types of sepsis diagnostic biomarkers are used in actual clinical practice rather than at the research level. (e.g. procalcitonin, IL-6, presepsin, etc.) This is because no biomarkers satisfy the sepsis trajectory requiring rapid diagnosis and treatment. Therefore, in the clinical field, there is no choice but to use the SOFA, qSOFA, and APACHE scores, which are the clinical signs or symptoms and the general blood test results. In particular, qSOFA is a fast assessment tool made statically through extensive cohort data and is not currently recommended for sepsis diagnosis. This study aimed to develop a scoring system that can be used in clinical practice through clinically applicable tests, which hypothesis is making the diagnosis of sepsis more accurate and faster in real-world practice. We also profoundly agree with you that BADS should be further scientifically verified through follow-up studies. Rather than replacing various scoring systems such as SOFA, APACHE, and SAPS score, we hope that the BADS will be studied as an additional scoring system that can help to diagnose sepsis at the bedside.

  • English writing should be polished and edited.

Response: We have edited with the English language again and attached the English writing proof of the ‘Editage.’

Reviewer 3 Report

I have no further comments

good

Author Response

July 2023

Editor, Journal of personalized medicine

Manuscript ID: jpm-2462528

Manuscript Title: Biomarker-based assessment model for detecting sepsis: A retrospective cohort study

We want to thank all of the editors and reviewers for helping us make a better revision. We revised our manuscript according to the comments and recommendations of the reviewers. Here we include a separate itemized series of responses to the comments of the reviewers.

----------------------------------------------------------------------------------------

Reviewer #2

  • I have no further comments

Response: Thank you very much. Your thoughtful peer review has helped us a lot to improve our research.

Round 3

Reviewer 1 Report

No other recommendations.

No other recommendations.